# Improved Frequency Estimation Algorithms with and without Predictions

**Anders Aamand**
MIT
aamand@mit.edu

**Justin Y. Chen**
MIT
justc@mit.edu

**Huy Lê Nguyễn**
Northeastern University
hu.nguyen@northeastern.edu

**Sandeep Silwal**
MIT
silwal@mit.edu

**Ali Vakilian**
TTIC
vakilian@ttic.edu

## Abstract

Estimating frequencies of elements appearing in a data stream is a key task in large-scale data analysis. Popular sketching approaches to this problem (e.g., CountMin and CountSketch) come with worst-case guarantees that probabilistically bound the error of the estimated frequencies for any possible input. The work of Hsu et al. (2019) introduced the idea of using machine learning to tailor sketching algorithms to the specific data distribution they are being run on. In particular, their learning-augmented frequency estimation algorithm uses a learned heavy-hitter oracle which predicts which elements will appear many times in the stream. We give a novel algorithm, which in some parameter regimes, already theoretically outperforms the learning based algorithm of Hsu et al. *without* the use of any predictions. Augmenting our algorithm with heavy-hitter predictions further reduces the error and improves upon the state of the art. Empirically, our algorithms achieve superior performance in all experiments compared to prior approaches.

## 1   Introduction

In frequency estimation, we stream a sequence of elements from $[n] := \{1, \ldots, n\}$, and the goal is to estimate $f_i$, the frequency of the $i$th element, at the end of the stream using low-space. Frequency estimation is one of the central problems in data streaming with a wide range of applications from networking (gathering important monitoring statistics [31, 62, 46]) to machine learning (NLP [33], feature selection [3], semi supervised learning [58]). CountMin (CM) [20] and CountSketch (CS) [14] are arguably the most popular and versatile of the algorithms for frequency estimation, and are implemented in many popular packages such as Spark [63], Twitter Algebird [10], and Redis.

Standard approaches to frequency estimation are designed to perform well in the worst-case due to the multitudinous benefits of worst-case guarantees. However, algorithms designed to handle any possible input do not exploit special structure of the particular distribution of inputs they are used for. In practice, these patterns can be described by domain experts or learned from historical data. Following the burgeoning trend of combining machine learning and classical algorithm design, [36] initiated the study of *learning-augmented* frequency estimation by extending the classical CM and CS algorithms in a simple but effective manner via a heavy-hitters oracle. During a training phase, they construct a classifier (e.g. a neural network) to detect whether an element $i$ is "heavy" (e.g., whether $f_i$ is among the most frequent items). After such a classifier is trained, they scan the input stream, and apply the classifier to each element $i$. If the element is predicted to be heavy, it is allocated a unique bucket, so that an exact value of $f_i$ is computed. Otherwise, the stream element is inputted into the standard sketching algorithms.

37th Conference on Neural Information Processing Systems (NeurIPS 2023).

The advantage of their algorithm was analyzed under the assumption that the true frequencies follow a heavy-tailed Zipfian distribution. This is a common and natural reoccurring pattern in real world data where there are a few very frequent elements and many infrequent elements. Experimentally, [36] showed several real datasets where the Zipfian assumption (approximately) held and useful heavy-hitter oracles could be trained in practice. Our paper is motivated by the following natural questions and goals in light of prior works:

> *Can we design better frequency estimation algorithms (with and without predictions) for heavy-tailed distributions?*

In particular, we consider the setting of [36] where the underlying data follow a heavy-tailed distribution and investigate whether sketching algorithms can be further tailored for such distributions. Before tackling this question, we must tightly characterize the benefits–and limitations–of these existing methods, which is another goal of our paper:

> *Give tight error guarantees for CountMin and CountSketch, as well as their learning-augmented variants, on Zipfian data.*

Lastly, any algorithms we design must possess worst case bounds in the case that either the data does not match our Zipfian (or more generally, heavy-tailed) assumption or the learned predictions have high error, leading to the following 'best of both worlds' goal:

> *Design algorithms which exploit heavy tailed distributions and ML predictions but also maintain worst-case guarantees.*

We addresses these challenges and goals and our contributions can be summarized as follows:

- We give tight upper and lower bounds for CM and CS, with and without predictions, for heavy tailed distributions. A surprising conclusion from our analysis is that (for a natural error metric) a constant number of rows is optimal for both CM and CS. In addition, our theoretical analysis shows that CS outperforms CM, both with and without predictions, validating the experimental results of [36].

- We go beyond CM and CS based algorithms to give a better frequency estimation algorithm for heavy tailed distributions, with and without the use of predictions. We show that our algorithms can deliver up to a logarithmic factor improvement in the error bound over CS and its learned variant. In addition, our algorithm has worst case guarantees.

- Prior learned approaches require querying an oracle for every element in the stream. In contrast, we obtain a *parsimonious* version of our algorithm which only requires a limited number of queries to the oracle. The number of queries we use is approximately equal to the given space budget.

- Lastly, we evaluate our algorithms on two real-world datasets with and without ML based predictions and show superior empirical performance compared to prior work in all cases.

## 1.1 Preliminaries

**Notation and Estimation Error**   The stream updates an $n$ dimensional frequency vector and every stream element is of the form $(i, \Delta)$ where $i \in [n]$ and $\Delta \in \mathbb{R}$ denotes the update on the coordinate. The final frequency vector is denoted as $f \in \mathbb{R}^n$. Let $N = \sum_{i \in [n]} f_i$ denote the sum of all frequencies. To simplify notation, we assume that $f_1 \geq f_2 \geq \ldots \geq f_n$. $\tilde{f}_i$ denotes the estimate of the frequency $f_i$. Given estimates $\{\tilde{f}_i\}_{i \in [n]}$, the error of a particular frequency is $|\tilde{f}_i - f_i|$. We also consider the following notion of overall weighted error as done in [36]:

$$\text{Weighted Error:} = \frac{1}{N} \sum_{i \in [n]} f_i \cdot |\tilde{f}_i - f_i|. \tag{1}$$

The weighted error can be interpreted as measuring the error with respect to a query distribution which is the same as the actual frequency distribution. As stated in [36], theoretical guarantees of frequency estimation algorithms are typically phrased in the traditional $(\varepsilon, \delta)$-error formulations. However as argued in there, the simple weighted objective (1) is a more holistic measure and does not require tuning of two different parameters, and is thus more natural from an ML perspective.

**Zipfian Stream**  We also work under the common assumption that the frequencies follow the Zipfian law, i.e., the $i$th largest frequency $f_i$ is equal to $A/i$ for some parameter $A$. Note we know $A$ at the end of the stream since the stream length is $A \cdot H_n$. By rescaling, we may assume that $A = 1$ without loss of generality. We will make this assumption throughout the paper.

**CountMin (CM)**  For parameters $k$ and $B$, which determine the total space used, CM uses $k$ independent and uniformly random hash functions $h_1, \ldots, h_k : [n] \to [B]$. Letting $C$ be an array of size $[k] \times [B]$ we let $C[\ell, b] = \sum_{j \in [n]}[h_\ell(j) = b]f_j$. When querying $i \in [n]$ the algorithm returns $\tilde{f}_i = \min_{\ell \in [k]} C[\ell, h_\ell(i)]$. Note that we always have that $\tilde{f}_i \geq f_i$.

**CountSketch (CS)**  In CS, we again have the hash functions $h_i$ as above as well as sign functions $s_1, \ldots, s_k : [n] \to \{-1, 1\}$. The array $C$ of size $[k] \times [B]$ is now tracks $C[\ell, b] = \sum_{j \in [n]}[h_\ell(j) = b]s_\ell(j)f_j$. When querying $i \in [n]$ the algorithm returns the estimate $\tilde{f}_i = \text{median}_{\ell \in [k]} s_\ell(i) \cdot C[\ell, h_\ell(i)]$.

**Learning-Augmented Sketches [36]**  Given a base sketching algorithm (either CM or CS) and a space budget $B$, the corresponding learning-augmented algorithm (learned CM or learned CS) allocates a constant fraction of the space $B$ to the base sketching algorithm and the rest of the space to store items identified as heavy by a learned predictor. These items predicted to be heavy-hitters are stored in a separate table which maintains their counts exactly, and their updates are not sent to the sketching algorithm.

## 1.2  Summary of Main Results and Paper Outline

Our analysis, both of CM and CS, our algorithm, and prior work, is summarized in Table 1.

| Algorithm | Weighted Error | Uses Predictions? | Reference |
|---|---|---|---|
| CountMin (CM) | $\Theta\left(\frac{\log n}{B}\right)$ | No | Theorem B.1 |
| CountSketch (CS) | $\Theta\left(\frac{1}{B}\right)$ | No | Theorem C.4 |
| Learned CountMin | $\Theta\left(\frac{\log(n/B)^2}{B \log n}\right)$ | Yes | [36] |
| Learned CountSketch | $\Theta\left(\frac{\log(n/B)}{B \log n}\right)$ | Yes | Theorem D.1 |
| Our (Without predictions) | $O\left(\frac{\log B + \text{poly}(\log \log n)}{B \log n}\right)$ | No | Theorem 2.1 |
| Our (Learned version) | $O\left(\frac{1}{B \log n}\right)$ | Yes | Theorem 3.1 |

Table 1: Bounds are stated assuming that the total space is $B$ words of memory. Weighted error means that element $i$ is queried with probability proportional to $1/i$. Moreover, the table considers normalized frequencies, so that $f_i = 1/i$.

**Summary of Theoretical Results**  We interpret Table 1. $B$ denotes the space bound, which is the total number of entries used in the CM or CS tables. First note that CS achieves lower weighted error compared to CM, proving the empirical advantage observed in [36]. However, the learned version of CS only improves upon standard CS in the regime $B = n^{1-o(1)}$. While this setting does appear sometimes in practice [33, 36] (referred to as high-accuracy regime), for CS, learning gives no asymptotic advantage in the low space regime.

On the other hand, in the low space regime of $B = \text{poly}(\log n)$, our algorithm, without predictions, already archives close to a logarithmic factor improvement over even *learned* CS. Furthermore, our learning-augmented algorithm achieves a logarithmic factor improvement over classical CS across all space regimes, whereas the learned CS only achieves a logarithmic factor improvement in the regime $B = n^{1-o(1)}$. Furthermore, our learned version outperforms or matches learned CS in all space regimes.

Our learning-augmented algorithm can also be made *parsimonious* in the sense that we only query the heavy-hitter oracle $\tilde{O}(B)$ times. This is desirable in large-scale streaming applications where evaluating even a small neural network on every single element would be prohibitive.

**Remark 1.1.** *We remark that all bounds in this paper are proved by bounding the expected error when estimating the frequency of a single item, $\mathbb{E}[|\tilde{f}_i - f_i|]$, then using linearity of expectation. While we specialized our bounds to a query distribution which is proportional to the actual frequencies in (1), our bounds can be easily generalized to any query distribution by simply weighing the expected errors of different items according to the given query distribution.*

**Summary of Empirical Results**   We compare our algorithm without prediction to CS and our algorithm with predictions to that of [36] on synthetic Zipfian data and on two real datasets corresponding to network traffic and internet search queries. In all cases, our algorithms outperform the baselines and often by a significant margin (up to **17x** in one setting). The improvement is especially pronounced when the space budget is small.

**Outline of the Paper**   Our paper is divided into roughly two parts. One part covers novel and tight analysis of the classical algorithms CountMin (CM) and CountSketch (CS). The second part covers our novel algorithmic contributions which go beyond CM and CS. The main body of our paper focuses on our novel algorithmic components, i.e. the second part, and we defer our analysis of the performance of CountMin (CM) and CountSketch (CS), with and without predictions, to the appendix: in Section B we give tight analysis of CM for a Zipfian frequency distribution. In Section C we give the analogous bounds for CS. Lastly, Section D gives tight bounds for CS with predictions. Section 2 covers our better frequency estimation without predictions while Section 3 covers the learning-augmented version of the algorithm, as well as its extentions.

### 1.3   Related Works

**Frequency Estimation**   While there exist other frequency estimation algorithms beyond CM and CS (such as [51, 48, 21, 40, 49, 11] ) we study hashing based methods such as CM [20] and CS [14] as they are widely employed in practice and have additional benefits, such as supporting insertions *and deletions*, and have applications beyond frequency estimation, such as in machine learning (feature selection [3], compressed sending [13, 25], and dimensionality reduction [61, 18] etc.).

**Learning-augmented algorithms**   The last few years have witnessed a rapid growth in using machine learning methods to improve "classical" algorithmic problems. For example, they have been used to improve the performance of data structures [42, 52], online algorithms [47, 56, 32, 5, 60, 43, 1, 6, 4, 22, 34], combinatorial optimization [41, 7, 43, 53, 23, 16], similarity search and clustering [59, 24, 30, 54, 57]. Similar to our work, sublinear constraints, such as memory or sample complexity, have also been studied under this framework [36, 38, 39, 19, 27, 28, 15, 44, 57].

## 2   Improved Algorithm without Predictions

We first present our frequency estimation algorithm which does not use any predictions. Later, we build on top of it for our final learning-augmented frequency estimation algorithm.

The main guarantees of of the algorithm is the following:

**Theorem 2.1.** *Consider Algorithm 1 with space parameter $B \geq \log n$ updated over a Zipfian stream. Let $\{\hat{f}_i\}_{i=1}^n$ denote the estimates computed by Algorithm 2. The expected weighted error (1) is $\mathbb{E}\left[ \frac{1}{N} \cdot \sum_{i=1}^n f_i \cdot |f_i - \hat{f}_i| \right] = O\left( \frac{\log B + poly(\log \log n)}{B \log n} \right)$.*

**Algorithm and Proof intuition:**   Let $B' = B/\log \log n$. At a high level, we show that for every $i \leq B'$, we execute line 10 of Algorithm 2 and the error satisfies $|1/i - \hat{f}_i| \approx 1/B'$ (recall in the Zipfian case, the $i$th largest frequency is $f_i = 1/i$). On the other hand, for $i \geq B'$, we show that (with sufficiently high probability) line 8 of Algorithm 2 will be executed, resulting in $|1/i - \hat{f}_i| = |1/i - 0| = 1/i$.

---

**Algorithm 1** (Not augmented) Frequency update algorithm

---

1: **Input:** Stream of updates to an $n$ dimensional vector, space budget $B$
2: **procedure** UPDATE
3:     $T \leftarrow \Theta(\log \log n)$
4:     **for** $j = 1$ to $T - 1$ **do**
5:         $S_j \leftarrow$ CountSketch table with 3 rows and $\frac{B}{6T}$ columns
6:     **end for**
7:     $S_T \leftarrow$ CountSketch table with 3 rows and $\frac{B}{6}$ columns
8:     **for** stream element $(i, \Delta)$ **do**
9:         Input $(i, \Delta)$ in each of the $T$ CountSketch tables $S_j$
10:     **end for**
11: **end procedure**

---

---

**Algorithm 2** (Not augmented) Frequency estimation algorithm

---

1: **Input:** Index $i \in [n]$ for which we want to estimate $f_i$
2: **procedure** QUERY
3:     **for** $j = 1$ to $T - 1$ **do**
4:         $\hat{f}_i^j \leftarrow$ estimate of the $i$th frequency given by table $S_j$
5:     **end for**
6:     $\tilde{f}_i \leftarrow \text{Median}(\hat{f}_i^1, \ldots, \hat{f}_i^{T-1})$
7:     **if** $\tilde{f}_i < O((\log \log n))/B$ **then**
8:         **Return** 0
9:     **else**
10:         **Return** $\hat{f}_i^T$, the estimate given by table $S_T$
11:     **end if**
12: **end procedure**

---

It might be perplexing at first sight why we wish to set the estimate to be $0$, but this idea has solid intuition: it turns out the *additive* error of standard CountSketch with $B'$ columns is actually of the order $1/B'$. Thus, it does not make sense to estimate elements whose true frequencies are much smaller than $1/B'$ using CountSketch. A challenge is that we do not know a priori which elements these are. We circumvent this via the following reasoning: if CountSketch itself outputs $\approx 1/B'$ as the estimate, then either one of the following must hold:

- The element has frequency $1/i \ll 1/B'$, in which case we should set the estimate to $0$ to obtain error $1/i$, as opposed to error $1/B' - 1/i \approx 1/B'$.

- The true element has frequency $\approx 1/B'$ in which case either using the output of the CountSketch table or setting the estimate to $0$ both obtain error approximately $O(1/B')$, so our choice is inconsequential.

In summary, the output of CountSketch itself suggests whether we should output an estimated frequency as $0$. We slightly modify the above approach with $O(\log \log n)$ repetitions to obtain sufficiently strong concentration, leading to a *robust* method to identify small frequencies. The proof formalizes the above plan and is given in full detail in Section E.

By combining our algorithm with predictions, we obtain improved guarantees.

## 3   Improved Learning-Augmented Algorithm

**Theorem 3.1.** *Consider Algorithm 3 with space parameter $B \geq \log n$ updated over a Zipfian stream. Suppose we have access to a heavy-hitter oracle which correctly identifies the top $B/2$ heavy-hitters in the stream. Let $\{\hat{f}_i\}_{i=1}^n$ denote the estimates computed by Algorithm 4. The expected weighted error (1) is $\mathbb{E}\left[\frac{1}{N} \cdot \sum_{i=1}^n f_i \cdot |f_i - \hat{f}_i|\right] = O\left(\frac{1}{B \log n}\right)$.*

---

**Algorithm 3** (Learning-augmented) Frequency update algorithm

---

1: **Input:** Stream of updates to an $n$ dimensional vector, space budget $B$, access to a heavy-hitter oracle which correctly identifies the top $B/2$ heavy-hitters
2: **procedure** UPDATE
3:     $T \leftarrow O(\log \log n)$
4:     **for** $j = 1$ to $T - 1$ **do**
5:         $S_j \leftarrow$ CountSketch table with 3 rows and $\frac{B}{12T}$ columns
6:     **end for**
7:     $S_T \leftarrow$ CountSketch table with 3 rows and $\frac{B}{12}$ columns
8:     **for** stream element $(i, \Delta)$ **do**
9:         **if** $i$ is a top $B/2$ heavy-hitter **then**
10:             Maintain the frequency of $i$ exactly
11:         **else**
12:             Input $(i, \Delta)$ in each of the $T$ CountSketch tables $S_j$
13:         **end if**
14:     **end for**
15: **end procedure**

---

---

**Algorithm 4** (Learning-augmented) Frequency estimation algorithm

---

1: **Input:** Index $i \in [n]$ for which we want to estimate $f_i$
2: **procedure** QUERY
3:     **if** $i$ is a top $B/2$ heavy-hitter **then**
4:         Output the exact maintained frequency of $i$
5:     **else**
6:         **Return** $\hat{f}_i \leftarrow$ output of Alg. 2 using the CountSkech tables created in Alg.3
7:     **end if**
8: **end procedure**

---

**Algorithm and Proof Intuition:** Our final algorithm follows a similar high-level design pattern used in the learned CM algorithm of [36]: given an oracle prediction, we either store the frequency of heavy element directly, or input the element into our algorithm from the prior section which does not use any predictions.

The workhorse of our analysis is the proof of Theorem 2.1. First note that we obtain 0 error for $i < B/2$. Thus, all error comes from indices $i \geq B/2$. Recall the intuition for this case from Theorem 2.1: we want to output 0 as our estimates as this results in lower error than the additive error from CS. The same analysis as in the proof of Theorem 2.1 shows that we are able to detect small frequencies and appropriately output an estimate from either the $T$th CS table or output 0.

## 3.1 Parsimonious Learning

In Theorem 3.1, we assumed access to a heavy-hitter oracle which we can use on every single stream element to predict if it is heavy. In practical streaming applications, this will likely be infeasible. Indeed, even if the oracle is a small neural network, it is unlikely that we can query it for every single element in a large-scale streaming application. We therefore consider the so called *parsimonious* setting with the goal of obtaining the same error bounds on the expected error but with an algorithm that makes *limited queries* to the heavy-hitter oracle. This setting has recently been explored for other problems in the learning-augmented literature [37, 9, 26].

Our algorithm works similarly to Algorithm 3 except that when an element $(i, \Delta)$ arrives, we only query the heavy-hitter oracle with some probability $p$ (proportional to $\Delta$). We will choose $p$ so that we in expectation only query $\tilde{O}(B)$ elements, rather than querying the entire stream. To be precise, whenever an item arrives, we first check if it is already classified as one of the top $B/2$ heavy-hitters in which case, we update its exact count (from the point in time where was classified as heavy). Otherwise, we query the heavy-hitter oracle with probability $p$. In case the item is queried and is indeed one of the top $B/2$ heavy-hitters, we start an exact count of that item. An arriving item which

is not used as a query for the heavy-hitter oracle and was not earlier classified as a heavy-hitter is processed as in Algorithm 3.

Querying for an element, we first check if it is classified as a heavy-hitter and if so, we use the estimate from the separate lookup table. If not, we estimate its frequency using Algorithm 4. With this algorithm, the count of a heavy-hitter will be underestimated since it may appear several times in the stream before it is used as a query for the oracle and we start counting it exactly. However, with our choice of sampling probability, with high probability it will be sampled sufficiently early to not affect its final count too much. We present the pseudocode of the algorithm as well as the precise result and its proof in Appendix G.

## 3.2 Algorithm variant with worst case guarantees

In this section we discuss a variant of our algorithm with worst case guarantees. To be more precise, we consider the case where the actual frequency distribution is not Zipfian. The algorithm we discuss is actually a more general case of Algorithm 2 and in fact, it completely recovers the asymptotic error guarantees of Theorem 2.1 (as well as Theorem 4 if we use predictions).

Recall that Algorithm 2 outputs 0 when the estimated frequency is below $T/B$ for $T = O(\log \log n)$. This parameter has been tuned to the Zipfian case. As stated in Section 2, the main intuition for this parameter is that it is of the same order as the additive error inherent in CountSketch, which we discuss now. Denote by $f_{\overline{P}}$ the frequency vector where we zero out the largest $P$ coordinates. For every frequency, the expected additive error incurred by a CountSketch table with $B'$ columns is $O(\|f_{\overline{B'}}\|_2 / \sqrt{B'})$. In the Zipfian case, this is equal to $O\left(\frac{\|f_{\overline{B'}}\|_2}{\sqrt{B'}}\right) = O\left(\frac{1}{B'}\right)$, which is exactly the threshold we set[1]. Thus, our robust variant simply replaces this tuned parameter $O(T/B)$ with an estimate of $O(\|f_{\overline{B'}}\|_2 / \sqrt{B'})$ where $B' = B/T$. We given an algorithm which efficiently estimates this quantity in a stream. Note this quantity is only needed for the query phase.

**Lemma 3.2.** *With probability at least* $1 - \exp\left(\Omega\left(B\right)\right)$, *Algorithm 6 outputs an estimate $V$ satisfying*

$$\Omega\left(\|f_{\overline{3B'}}\|_2^2 / B'\right) \leq V \leq O\left(\left\|f_{\overline{B'/10}}\right\|_2^2 / B'\right).$$

The algorithm and analysis are given in Section H. Replacing the threshold in Line 7 of Algorithm 2 with the output of Algorithm 6 (more precisely the square root of the value) readily gives us the following worst case guarantees. Lemma 3.3 states that the expected error of the estimates outputted by Algorithm 2 using $B$, regardless of the true frequency distribution, is no worse than that of a standard CountSketch table using slightly smaller $O(B/\log \log n)$ space.

**Lemma 3.3.** *Suppose $B \geq \log n$. Let $\{\hat{f}_i\}_{i=1}^n$ denote the estimates of Algorithm 2 using $B/2$ space and with Line 7 replaced by the square root of the estimate of Algorithm 6, also using $B/2$ space. Suppose the condition of Lemma 3.2 holds. Let $\{\hat{f}_i'\}_{i=1}^n$ denote the estates computed by a CountSketch table with $\frac{cB}{\log \log n}$ columns for a sufficiently small constant $c$. Then,* $\mathbb{E}[|\hat{f}_i - f_i|] \leq \mathbb{E}[|\hat{f}_i' - f_i|]$.

**Remark 3.1.** *The learned version of the algorithm automatically inherits any worst case guarantees from the unlearned (without predictions) version. This is because we only set aside half the space to explicitly track the frequency of some elements, which has worst case guarantees, while the other half is used for the unlearned version, also with worst case guarantees.*

## 4 Experiments

We experimentally evaluate our algorithms with and without predictions on real and synthetic datasets and demonstrate that the improvements predicted by theory hold in practice. Comprehensive additional figures are given in Appendix J.

**Algorithm Implementations** In the setting without predictions, we compare our algorithm to CountSketch (CS) (which was shown to have favorable empirical performance compared to CountMin (CM) in [36] and better theoretical performance due to our work). In the setting with predictions, we compare the algorithm of [36], using CS as the base sketch and dedicated half of the space for items

---

[1]Recall $B' = B/T$ in Algorithm 2.

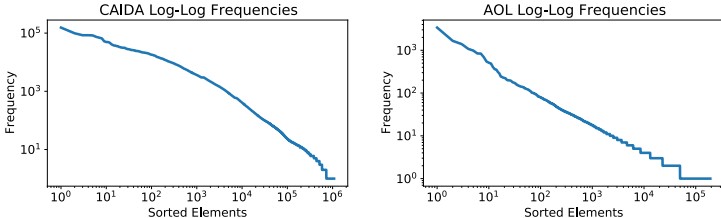

Figure 1: Log-log plots of the sorted frequencies of the first day/minute of the CAIDA/AOL datasets. Both data distributions are heavy-tailed with few items accounting for much of the total stream.

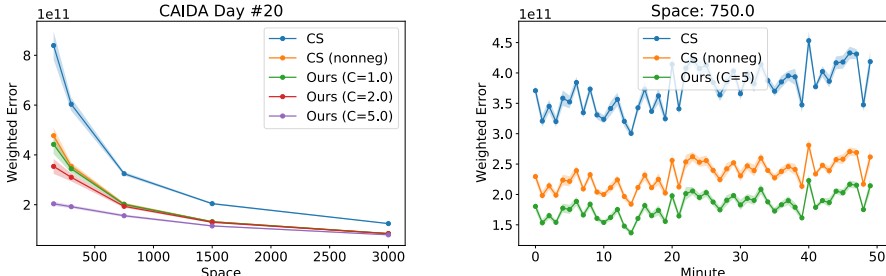

Figure 2: Comparison of weighted error without predictions on the CAIDA dataset. The left plot compares the performance of various algorithms (including our algorithm with different choices of $C$) for a fixed dataset and varying space. The right plot compares algorithms over time across separate streams for each minute of data for a specific choice of space being 750.

which are predicted to be heavy by the learned oracle. For all implementations, we use three rows in the CS table and vary the number of columns. We additionally augment each of these baselines with a version that truncates all negative estimated frequencies to zero as none of our datasets include stream deletions. This simple change does not change the asymptotic $(\varepsilon, \delta)$ classic sketching guarantees but does make a big difference when measuring empirical weighted error.

We implement a simplified and practical version of our algorithm which uses a single CS table. If the median estimate of an element is below a threshold of $Cn/w$ for domain size $n$, sketch width $w$ (a third of the total space), and a tunable constant $C$, the estimate is instead set to 0. As all algorithms use a single CS table as the basic building block with different estimation functions, for each trial we randomly sample hash functions for a single CS table and only vary the estimation procedure used.

We evaluate algorithms according the weighted error as in Equation (1) but also according to unweighted error which is simply the sum over all elements of the absolute estimation error, given by $\sum_i |f_i - \tilde{f}_i|$. Space is measured by the size of the sketch table, and all errors are averaged over 10 independent trials with standard deviations shown shaded in.

**Datasets** We compare our algorithm with prior work on three datasets. We use the same two real-world datasets and predictions from [36]: the CAIDA and AOL datasets. The CAIDA dataset [12] contains 50 minutes of internet traffic data. For each minute of data, the stream is formed of the IP addresses associated with packets going through a Tier1 ISP. A typical minute of data contains 30 million packets accounted for by 1 million IPs. The AOL dataset [55] contains 80 days of internet search queries with a typical day containing $\approx 3 \cdot 10^5$ total queries and $\approx 10^5$ unique queries. As shown in Figure 1, both datasets approximately follow a power law distribution. For both datasets, we use the predictions from prior work [36] formed using recurrent neural networks. We also generate synthetic data following a Zipfian distribution with $n = 10^7$ elements and where the $i$th element has frequency $n/i$.

**Results** Across the board, our algorithm outperforms the baselines. On the CAIDA and AOL datasets without predictions, our algorithm consistently outperforms the standard CS with up to **4x** smaller error with space 300. This gap widens when we compare our algorithm with predictions

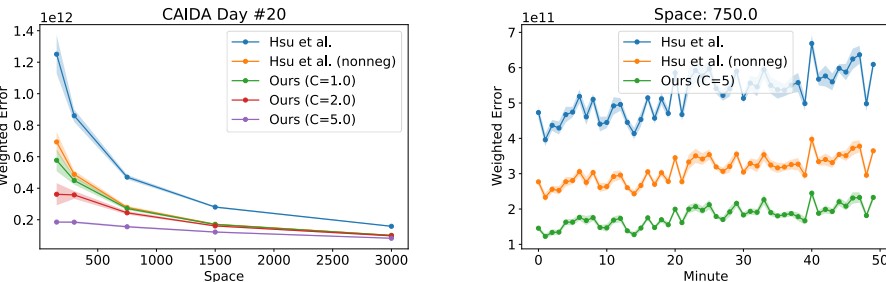

Figure 3: Comparison of weighted error with predictions on the CAIDA dataset.

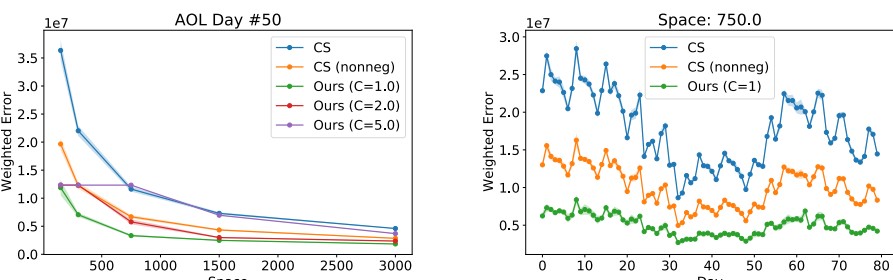

Figure 4: Comparison of weighted error without predictions on the AOL dataset.

to that of [36] with a gap of up to **17x** with space 300. In all cases, the performance of CS and [36] is significantly improved by the simple trick of truncating negative estimates to zero. However, our algorithm still outperforms these "nonneg" baselines. The longitudinal plots which compare algorithms over time show that our algorithm consistently outperforms the state-of-the-art with and without predictions.

In the case of the CAIDA dataset, predictions do not generally improve the performance of any of the algorithms. This is consistent with the findings of [36] where the prediction quality for the CAIDA dataset was relatively poor. However, for the AOL which has a more accurate learned oracle, our algorithm in particular is significantly improved when augmented with predictions. Intuitively, the benefit of our algorithm comes from removing error due to noise for low frequency elements. Conversely, good predictions help to obtain very good estimates of high frequency elements. In combination, this yields very small total weighted error.

In Appendix J, we display comprehensive experiments of the performance of the algorithms across the CAIDA and AOL datasets with varying space and for both weighted and unweighted error as well as results for synthetic Zipfian data. In all cases, our algorithm outperforms the baselines. On synthetic Zipfian, the gap between our algorithm and the non-negative CS for weighted error is relatively small compared to that for the real datasets. While we mainly focus on weighted error in this work, the benefits of our algorithm are even more significant for unweighted error as setting estimates below the noise floor to zero is especially impactful for this error measure. In general, we see the trend, matching our theoretical results, that as space increases, the gap between the different algorithms shrinks as the estimates of the base CS become more accurate.

## Acknowledgements

Anders Aamand is supported by DFF-International Postdoc Grant 0164-00022B from the Independent Research Fund Denmark and a Simons Investigator Award. Justin Chen is supported by an NSF Graduate Research Fellowship under Grant No. 174530. Huy Nguyen is supported by NSF Grants 2311649 and 1750716.

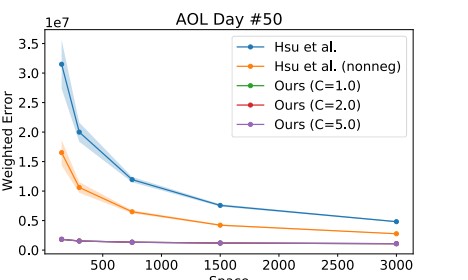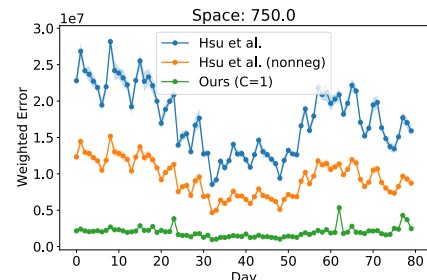

Figure 5: Comparison of weighted error with predictions on the AOL dataset.

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
