# OpenReview forum: "Improved Frequency Estimation Algorithms with and without Predictions"
_NeurIPS.cc/2023/Conference — NeurIPS 2023 spotlight_

### Official Review · Reviewer_iUr9 · 2023-06-28

**Soundness:** 3 good
**Presentation:** 2 fair
**Contribution:** 3 good
**Rating:** 6
**Confidence:** 3

**Summary:**

This paper studied frequency estimation and learning-augmented frequency estimation. CountMin and CountSketch are the most popular algorithms for this task. With the addition of learning augmentation, an algorithm is given access to a learned prediction, in this case the prediction of the heavy hitters. This paper focuses on the stream being from a Zipfian distribution, which are well-studied, well-motivated distributions with heavy tails.

In the learning-augmented algorithm, if an element is predicted to be heavy, it is given a unique bucket so that a more accurate frequency can be computed for it. If it isn’t predicted to be heavy, it is simply input into a sketching  algorithm. They prove bounds on the weighted error of algorithms, including,  CountSketch, CountMin, and a novel algorithm. For CountSketch and CountMin, the paper gives a tight analysis. The new algorithm is studied both with and without predictions, though predictions give the largest advantage in low space settings.

Experiments justify the theory is predictive of performance.

**Strengths:**

- Learning-augmented frequency estimation is itself a very nice question, I was looking forward to reading this paper in my pile.
- The algorithm is clean, straight-forward. I believe the results are correct.
- The paper is grammatically well-written.



**Weaknesses:**

- I am confused about the prediction model. Normally, in learning-augmented algorithms, we measure an algorithm’s performance based on the error in the prediction. Here, as far as I could tell, all of the theoretical results only held when one assumed the predicted heavy hitters were correct. I expected to see some trade-off between the quality of prediction and the weighted error bounds. The experiments briefly mentioned that the prediction quality might be poor, thus leading to worse empirical performance (as expected), but there was no theory discussing the robustness of the predictions. Robustness in the prediction error is what differentiates learning-augmented algorithm from all these other BWCA frameworks (data-driven algorithms, algorithms with advice, etc).
Perhaps because of the heavy tail distribution assumptions, it’s reasonable to assume that one learns the heavy hitters perfectly? Or can you offer another explanation for this choice in the model?

- This paper does not clearly lay out its improvements on prior work. I would like to see a lot more comparison to the most relevant previous work [Hsu et al. 2019]. Can this be more clearly stated  in the introduction? Concretely, it would help to have previously known results listed in a column in your table 1 for that we can see your improvement.

**Questions:**

(see Weaknesses, please)

---

> ### Author Rebuttal · Authors · 2023-08-10
>
> We thanks you for your thorough review and your comments. Below we address your questions and concerns.
>
> >I am confused about the prediction model. Normally, in learning-augmented algorithms, we measure an algorithm’s performance based on the error in the prediction. Here, as far as I could tell, all of the theoretical results only held when one assumed the predicted heavy hitters were correct.
>
> We would like to first point out that our algorithm without predictions (Theorem 2.1) already outperforms the learning-augmented version of the standard CountSketch algorithm supplied with perfect predictions, in low space regimes. Secondly, while it is true that the best bounds obtained in Theorem 3.1 assume access to perfect predictions, our learned version does indeed possess worst-case guarantees, even if *all* predictions are incorrect. The worst case guarantees follow simply by the design of the algorithm: it explicitly keeps track of the frequencies of a select number of elements (those deemed the top O(B) heavy elements by the predictor). For these elements we incur no error, even if they don’t turn out to be true heavy elements. For the other elements, our algorithm inputs them into our improved version of CountSketch without predictions, whose guarantees are listed in Theorem 2.1 (see Lemma 3.3 for a general version with no Zipfian assumptions). This type of worst-case guarantees as well as the prediction model is identical to prior works on learning-augmented algorithms for frequency estimation [1]. In addition, it mirrors the consistency/robustness guarantees that appear in several algorithms with predictions papers (see the survey [2]) which bound the performance with perfect predictions (consistency) and with arbitrary predictions (robustness).
> Lastly, we believe our best error guarantees, which are given in Theorem 3.1, can also be obtained with weaker prediction models. Our algorithm is more robust to false positives, meaning light elements which are classified as heavy, than false negatives, which are heavy elements classified as light. Therefore, it is likely that our “learned version” results extend to the case where the prediction’s accuracy is tied to the true heaviness of the frequency. Nevertheless, as demonstrated by our superior empirical results, our algorithm generalizes to noisy real world predictions. This demonstrates that the prediction model (used in our work and [1]) is a useful model to develop novel algorithms.
>
> >I expected to see some trade-off between the quality of prediction and the weighted error bounds. The experiments briefly mentioned that the prediction quality might be poor, thus leading to worse empirical performance (as expected), but there was no theory discussing the robustness of the predictions. Robustness in the prediction error is what differentiates learning-augmented algorithm from all these other BWCA frameworks (data-driven algorithms, algorithms with advice, etc). Perhaps because of the heavy tail distribution assumptions, it’s reasonable to assume that one learns the heavy hitters perfectly? Or can you offer another explanation for this choice in the model?
>
> We can in fact provide some trade-offs between the quality of the prediction and the weighted error bounds, specifically for the (Learned) CountSketch algorithm. We left these out of the submission but will consider including them in the paper. For expected error, these results give a smooth trade-off between the bounds for the classic and corresponding learned algorithms (row 2 and 4 of Table 1). The result assume that the predictor may misclassify an element with some probability $\delta$. If $\delta=O(\ln(n/B)/\ln(n))$, then it turns out that we asymptotically obtain the same expected error as with the learned variant which has access to perfect predictions. For the (Learned) CountMin algorithm (row 1 and 3 of Table 1), a similar trade-off is presented in [1] where it is also the case that when $\delta=O(\ln(n/B)/\ln(n))$, the learned variant of the algorithm has the same asymptotic expected error as with a perfect predictor. As indicated above, there is more interesting work to be done on the prediction error. Especially exploring the asymmetry in the robustness to respectively false positives and negatives is a direction for future work.
>
> >This paper does not clearly lay out its improvements on prior work. I would like to see a lot more comparison to the most relevant previous work [Hsu et al. 2019]. Can this be more clearly stated in the introduction? Concretely, it would help to have previously known results listed in a column in your table 1 for that we can see your improvement.
>
> The suggested table is already in the appendix and following the reviewer’s suggestion, we will move Table 2 in the main body and integrate it in Table 1. To summarize, [Hsu et al. 2019] only analyzed CM and its learned variant.
>
>
>
> [1] Chen-Yu Hsu, Piotr Indyk, Dina Katabi, and Ali Vakilian. Learning-based frequency estimation algorithms. ICLR 2019.
>
> [2] Michael Mitzenmacher and Sergei Vassilvitskii. 2022. Algorithms with predictions. Commun. ACM 65, 7 (July 2022), 33–35. https://doi.org/10.1145/3528087

---

> > ### Comment · Reviewer_iUr9 · 2023-08-15
> >
> > Authors: thank you very much for the thoughtful response. I apologize for the delay in this reply; I will be much prompter to continue discussion for the rest of the response period, if needed.
> >
> > Your responses more than adequately addressed my concerns about the trade-offs between the quality of the prediction and the weighted error bounds and my confusion about the definition of the quality of the prediction. I would encourage you to include your first two responses to me in the paper, if there's room. For someone very familiar with learning-augmented algorithms, but less-so with sketching problems,  these worst case/ robustness guarantees that you obtain (though pretty simple to explain!) were not obvious to me. And obviously they heavily impacted my understanding of the paper's contribution in the algorithms its predictions space.
> >
> > I will be updating my score accordingly.

---

> ### Author Response · Authors · 2023-08-13
> **Update check**
>
> Dear Reviewer iUr9,
>
> Did we address all your concerns satisfactorily, in particular your comments about the prediction model and our improvements over prior works? If your concerns have not been resolved, could you please let us know which concerns were not sufficiently addressed so that we have a chance to respond before the deadline?
>
> Many thanks, The authors

---

### Official Review · Reviewer_UWwz · 2023-07-06

**Soundness:** 3 good
**Presentation:** 4 excellent
**Contribution:** 4 excellent
**Rating:** 7
**Confidence:** 3

**Summary:**

Summary of the Paper
==================
* This work follows (Hsu Indyk Katabi Vakilian 2019) in trying to improve the performance of hashing-based frequency estimation algorithms (such as Count-Min, CountSketch) by making use of "advice" in the form of a learning model's predictions which classify the input elements as "heavy-hitters" or otherwise based on the input distribution.
* Just as (HIKV2019), the theoretical analysis assumes a Zipfian (heavy-tail) property for the data distribution, and they provide guarantees for the expected weighted estimation error $\frac{1}{N} \sum_{i=1}^{n} f_i \cdot |f_i - \hat{f}_i|$.
* They improve on the (HIKV2019) analysis of Count-Min and Learned-Count-Min algorithms to get tight bounds on the expected estimation error when there are multiple hash functions ($k \geq 2$).
 * They also provide tight bounds for the expected estimation error of CountSketch, with and without learning.
* Finally, they propose a better frequency estimation algorithm --- both plain (Algorithm 1&2) and learning-augmented (Algorithm 3&4) --- and prove bounds on the expected estimation error in both cases, showing that the learning-augmented algorithm outperforms both Plain-CS and Learned-CS in all regimes, whereas the plain (no-learning) algorithm outperforms the Plain-CS algorithm in the low-space regime ($B = {\rm polylog}(n)$).
* They also propose a parsimonious variant of the algorithm (limited number of queries) and do an experimental evaluation.

**Strengths:**


* The problem setting is already studied in the literature and thus the improvements shown in this work are clearer. The Zipfian (heavy-tail) property for the data distribution is known to hold for many real world datasets (if approximately).
* This work provides tight bounds for the expected estimation error of CountSketch and CountMin, both with and without learning. In the case of CountMin, it improves upon the existing bounds from (HIKV2019).
* The proposed "better frequency estimation algorithm" provides tangible improvements over CS and CM, both wiith and without learning-augmentation.
* They also consider a variation of the algorithm with worst-case guarantees, even when the data distribution is not Zipfian, and the variant nicely generalises from the Zipfian case.
* The work includes the implementation of the algorithms and experimental evaluation.
* A reasonable level of proof-sketches are provided in the main paper.

**Weaknesses:**

* The experiments should ideally have also considered the worst-case variant of the algorithm (Algorithm 6 in the supplementary) in both the Zipfian and non-Zipfian cases.

**Questions:**

None

**Limitations:**

Not applicable.

---

> ### Author Rebuttal · Authors · 2023-08-10
>
> We are happy to hear that you found our paper interesting and thank you for your time and comments.

---

### Official Review · Reviewer_TyeE · 2023-07-06

**Soundness:** 3 good
**Presentation:** 3 good
**Contribution:** 3 good
**Rating:** 7
**Confidence:** 3

**Summary:**

Authors study frequency estimation algorithms CountMin and CountSketch
and propose their modifications tailored for heavy tailed distributions.
They first analyze CountMin and CountSketch, showing that the second
one achieves better theoretical bounds on such distributions which explains
experimental results in previous work.
They propose a different algorithm with significantly better performance
bounds on heavy tailed distributions which also satisfies worst case guarantees
(for the case when the input does come from a considered heavy-tailed distribution)
which are comparable to CountSketch.
They also propose an ML-augmented variant of their algorithm which assumes that
there is an oracle which correctly identifies half of the heavy hitters. This algorithm
also works in parsimonious setting where it is allowed to receive only a few predictions.


**Strengths:**

* They consider problem important both in theory and practice in a setting which occurs often in practice
* They show limitations of the existing algorithms and design new ones overcoming these limitations
* the ML-augmented version of their algorithm can work in a parsimonious regime: only very few predictions are needed and I believe that this is a good sign of usability in practice

**Weaknesses:**

* I did not see lower bounds for the problem in their setting. It is not clear whether better algorithms are possible
* It is not clear how their algorithm's performance depend on precision of the predictor, e.g., what if it identifies too many or too few items as heavy hitters

**Questions:**

 * if your algorithm reports too many items as heavy hitters, what does your algorithm do?
* requirement that the predictor perfectly identifies the top B/2 heavy hitters seems rather strict. Can it be made weaker, e.g. that it identifies 90% of the top B heavy hitters, or that it correctly identifies $i$th heavy hitters with some probability depending on $i$?

**Limitations:**

assumptions clearly stated in the theoretical results

---

> ### Author Rebuttal · Authors · 2023-08-10
>
> >I did not see lower bounds for the problem in their setting. It is not clear whether better algorithms are possible
>
> Proving lower bounds for learning-augmented frequency estimation, or even frequency estimation under our expected error metric, is an interesting future research direction.
>
> >It is not clear how their algorithm's performance depend on precision of the predictor, e.g., what if it identifies too many or too few items as heavy hitters
>
> Our learned version has worst case guarantees, even if the predictions are totally incorrect. This is because the algorithm explicitly keeps track of the frequencies of a select number of elements (those deemed the top O(B) heavy elements by the predictor). For these elements we incur no error, even if they don’t turn out to be true heavy elements. For the other elements, our algorithm inputs them into our improved version of CountSketch without predictions, whose guarantees are listed in Theorem 2.1 (see Lemma 3.3 for a general version with no Zipfian assumptions). This is the same type of worst-case guarantees given by prior works such as Hsu et al.
>
>
> >If your algorithm reports too many items as heavy hitters, what does your algorithm do? The requirement that the predictor perfectly identifies the top B/2 heavy hitters seems rather strict. Can it be made weaker, e.g. that it identifies 90% of the top B heavy hitters, or that it correctly identifies ith heavy hitters with some  probability depending on i?
>
> For our best error bounds given in Theorem 3.1, we do assume that the predictor correctly identifies the top O(B) heavy elements. This is the same prediction model used in the prior work of Hsu et al. It is indeed likely that the prediction model can be relaxed to obtain similar improvements. Our algorithm is more robust to false positives, meaning light elements which are classified as heavy, than false negatives, which are heavy elements classified as light. Therefore, it is likely that our best results extend to the case where the prediction’s accuracy is tied to the true heaviness of the frequency. Nevertheless, as demonstrated by our superior empirical results, our algorithm generalizes to noisy real world predictions, demonstrating the versatility of the prediction model used in our work and Hsu et al.

---

> > ### Comment · Reviewer_TyeE · 2023-08-12
> >
> > thank you for your explanation.

---

### Official Review · Reviewer_5HgB · 2023-07-07

**Soundness:** 3 good
**Presentation:** 3 good
**Contribution:** 3 good
**Rating:** 7
**Confidence:** 4

**Summary:**

The authors present a new error analysis for Count-Sketch (CS) and Count-Min Sketch (CMS) for heavy-tailed distributions. They propose a novel Count-Sketch-based algorithm and its learned variant to estimate the frequencies of items in a data stream. Empirically, they show that both algorithms outperform the standard CS and LCMS of Hsu et al. (ICLR 2019) in terms of weighted and unweighted estimation errors on various datasets. In addition, they introduce a parsimonious version of their learning-based algorithm which performs a limited number of queries to the oracle.

**Strengths:**

The paper is concerned with the fundamental problem of estimating the frequencies of items in a data stream. It introduces tight error guarantees for Count-Min sketch and Count-Sketch algorithms as well as their learned variants for Zipfian distributions. The authors propose a novel Count-Sketch-based algorithm and its learning-augmented variant that significantly outperform the baseline algorithms on two real-world datasets and a synthetic Zipfian dataset.

**Weaknesses:**

The results section of the paper mentions that the prediction quality for the CAIDA dataset was relatively poor, however, the work of Hsu et al. (ICLR 2019) states that the AUC score of identifying the top 1% heavy hitters for CAIDA was 0.1 higher than for the AOL dataset. Hence, it seems that the experimental results are not quite consistent with those of Hsu et al. as their LCMS offered more significant advantages than the basic CMS on the CAIDA dataset as compared to AOL.

Furthermore, the experimental section does not include results for the parsimonious algorithm and less heavy-tailed distributions.

**Questions:**

The learned models for the CAIDA and AOL datasets are very complex and rather expensive to train, and the space allocation of the learned variant in the given plots does not seem to include the space reserved for the oracle. Therefore, it is unclear whether the novel learning-based algorithm offers any advantages over using its non-learned counterpart. It would be great to have a comparison of these two algorithms with equal space allocations which includes the space for the learned model.

In addition, it would be helpful if the accuracies of identifying top B/2 frequent items of the learned oracles for the AOL and CAIDA datasets were given in the paper as well as a comparison of the B/2 value to the thresholds used in LCMS of Hsu et al. (ICLR 2019) to investigate why the learning-based variant of the novel CS algorithm does not offer similar advantages for the CAIDA dataset as LCMS.

**Limitations:**

It would be helpful if the limitation of the parsimonious algorithm due to having to estimate the length of the data stream was stated more clearly in the paper as well as that of Algorithm 2 for non-heavy-tailed distributions due to having to estimate the tail of the frequency vector.

---

> ### Author Rebuttal · Authors · 2023-08-10
>
> We thank you for your interest in our paper and your comments. We address questions and concerns below
>
> >It appears that for non-Zipfian distributions, the non-simplified Algorithm 2 would have to perform two passes over the data stream since Algorithm 6 would need to first output an estimate of the L2 norm of the tail of the frequency vector which presents a major drawback.
>
> We believe there is a misunderstanding and we hope our comment clarifies it. All of our algorithms only require \emph{one pass} over the stream. Algorithm 2 outputs an estimate of the frequencies \emph{after} the stream has ended. Furthermore, the output of Algorithm 6 is only used to compute approximate frequencies after the stream has ended. Thus, both Algorithm 1, our main streaming algorithm, and Algorithm 6, which estimates the tail norm of the frequency stream, can be run in one pass in parallel. Later, after the stream ends, they can be combined to output approximate estimates.
>
> >The results section of the paper mentions that the prediction quality for the CAIDA dataset was relatively poor, however, the work of Hsu et al. (ICLR 2019) states that the AUC score of identifying the top 1% heavy hitters for CAIDA was 0.1 higher than for the AOL dataset. Hence, it seems that the experimental results are not quite consistent with those of Hsu et al. as their LCMS offered more significant advantages than the basic CMS on the CAIDA dataset as compared to AOL.
>
> Thanks for pointing out this discrepancy. Indeed, the predictions we have for CAIDA are worse than that of AOL (accuracy of recovering the top 1000 say is around 0.4 for AOL and 0.2 for CAIDA). We are working to investigate why this discrepancy exists and will update the paper accordingly as well as include quantitative details on the prediction quality.
>
> >The learned models for the CAIDA and AOL datasets are very complex and rather expensive to train, and the space allocation of the learned variant in the given plots does not seem to include the space reserved for the oracle. Therefore, it is unclear whether the novel learning-based algorithm offers any advantages over using its non-learned counterpart. It would be great to have a comparison of these two algorithms with equal space allocations which includes the space for the learned model.
>
> Accounting for the space used for learning is an important consideration. As pointed out in Hsu et al. in the setting where we are sketching many subsequent streams of data (e.g., different days), the space cost for storing a learned model can be amortized over time. We will add a brief discussion of this point to the paper including quantitative details.
>
> >In addition, it would be helpful if the accuracies of identifying top B/2 frequent items of the learned oracles for the AOL and CAIDA datasets were given in the paper as well as a comparison of the B/2 value to the thresholds used in LCMS of Hsu et al. (ICLR 2019) to investigate why the learning-based variant of the novel CS algorithm does not offer similar advantages for the CAIDA dataset as LCMS.
>
> See above.
>
> >It would be helpful if the limitation of the parsimonious algorithm due to having to estimate the length of the data stream was stated more clearly in the paper as well as that of Algorithm 2 for non-heavy-tailed distributions due to having to estimate the tail of the frequency vector.
>
> Thanks for the comment. We agree that it would be useful to have this stated more clearly in the paper, and will do so in the final version.

---

> > ### Comment · Reviewer_5HgB · 2023-08-21
> >
> > Thank you for the detailed clarification.

---

### Official Review · Reviewer_yxpU · 2023-07-12

**Soundness:** 4 excellent
**Presentation:** 4 excellent
**Contribution:** 4 excellent
**Rating:** 8
**Confidence:** 3

**Summary:**

The authors study frequency estimation in a streaming setting using CountMin and CountSketches, both their classic and learning augmented variants. They prove tight theoretical bounds for the expected error when the frequencies follow the Zipf distribution.
They also introduce and analyze a new algorithm with lower error that returns 0 for low frequencies instead of the noisy estimates of classic CountSketch. Furthermore they also introduce a parsimonious version of their algorithm that avoids consulting the potentially much slower machine learned model for each item of the stream using Poisson sampling to provably invoke it a small number of times only. Several experiments with two real world and synthetic data sets support the claims, albeit the implemented algorithm is much simpler than the one analyzed and a simple modification of the classic CountSketch also yields substantial improvements.

**Strengths:**

1) Problem and techniques studied are extremely well motivated and widely used.
2) Solid theoretical analysis and tight new lower and upper bounds.
3) Introduces multiple new algorithm variants.
4) Substantial error reduction in the experiments.
5) Paper is well written and structured.



**Weaknesses:**

1) No experiments with the theoretically analyzed algorithm, no theory for the simpler variant in the experiments.
2) I would love to see some experiments with the parsimonious algorithm as well.
3) When its truncation threshold is properly tuned the experimentally evaluated simplified algorithm is more accurate than returning max {0, CountSketch's estimate}. However the best threshold is dataset dependent and the wrong threshold underperforms the non-negative CountSketch (i.e. threshold = 0). Section 3.2 proposes a theoretical construction based on the Alon-Matias-Szegedy sketch to adaptively tune and set the threshold, nevertheless this variant is not evaluated in the experiments either. It would be good to evaluate a hyper-parameter free variant that works (well) on any data out of the box or explicitly leave it as future work.

**Questions:**

Alg 1: What's median of 4? The proof section carefully requires odd number of rows. Could you please clarify, or since it's only for the sake of theory make it 3 (or 5) to keep it simple?

Alg 2: Could you discuss why it's essential (or not) to take median of medians instead of using a single CountSketch with O(T) rows as a filter?

Could you also discuss whether your results hold (strengthen or weaken) for more general power laws where f_i ~ (1/i)^p (or log-normal) beyond f_i ~ 1/i Zipf similarly to Du, Elbert, Franklyn Wang, and Michael Mitzenmacher. "Putting the “Learning." ICML, 2021? Probably it's best worked out and discussed after lines 222-223.

Could you also measure and disclose the power law exponent for the CAIDA and AOL datasets?

Figures 2-5: Could you use the same color for best Our (C=..) line in the left and right sub-plots?

Lines 249-250: three columns and varying number of rows -> 3 rows and varying number of columns (typo).

**Limitations:**

Yes, it's absolutely forthcoming and adequate.

---

> ### Author Rebuttal · Authors · 2023-08-10
>
> We are glad to hear you found our paper interesting and appreciate your comments! We address them below:
>
> >No experiments with the theoretically analyzed algorithm, no theory for the simpler variant in the experiments.
>
> The specific setting of parameters for the theoretical algorithm (number of CS tables, threshold for heavy/light elements) are chosen to achieve the best asymptotic bounds we were able to prove. In practice, our belief (backed up by the experimental results) is that the core algorithmic idea of truncating low estimates to zero will yield benefits, but that the specific parameters/setup best for the asymptotics are likely not the best when you process a specific dataset. It is a very nice question whether a simplified algorithm could be shown to achieve the same bounds as our algorithm which uses $O(\log \log n)$ tables.
>
> > I would love to see some experiments with the parsimonious algorithm as well.
>
> We are glad to hear your interest and will try to include experiments using the parsimonious version of the learned algorithm in the final paper.
>
> >When its truncation threshold is properly tuned the experimentally evaluated simplified algorithm is more accurate than returning max {0, CountSketch's estimate}. However the best threshold is dataset dependent and the wrong threshold underperforms the non-negative CountSketch (i.e. threshold = 0). Section 3.2 proposes a theoretical construction based on the Alon-Matias-Szegedy sketch to adaptively tune and set the threshold, nevertheless this variant is not evaluated in the experiments either. It would be good to evaluate a hyper-parameter free variant that works (well) on any data out of the box or explicitly leave it as future work.
>
> This is a fair point, and we will make this explicit in the paper. At least for the learned algorithms, as the high-level idea for applications is that the user is processing similar data over time (and therefore can learn some structure), we believe it is reasonable to also think that they can tune hyperparameters on past data. Though this may not be the case in applications of the non-learned algorithms, and we will mention this.
>
> >Alg 1: What's median of 4? The proof section carefully requires odd number of rows. Could you please clarify, or since it's only for the sake of theory make it 3 (or 5) to keep it simple?
>
> Thank you for your comment and pointing out the typo. Indeed, the 4 should be a 3 (or any fixed odd integer at least 3). We will fix the typo in the updated version of the paper.
>
> > Alg 2: Could you discuss why it's essential (or not) to take median of medians instead of using a single CountSketch with $O(T)$ rows as a filter?
>
> This is a very crucial but subtle point and we thank you for pointing it out. We hope the following explanation is informative. First, we know that a *single* CountSketch (CS) table cannot achieve the guarantees of our best algorithm. Indeed, Table 1 (and our Theorem C.4) gives the tight error bound for a single CS table. The intuition for why a single CS table is not sufficient is roughly as follows: a large portion of the error is incurred due to elements whose true frequencies are much smaller than the expected error guarantees of CS (more rows only improve the concentration of the error, not the expected value!). Thus, we cannot simply rely on estimates from a CS table and must do something different.
> Indeed, our algorithm goes beyond CS by employing a two step procedure. The first step can be thought of as a filtering step like you stated. This filtering step informs us whether to use the estimate of the very last (large) CountSketch table or output 0. Multiple tables in the first step simply reduces the probability that we accidentally use the estimate of the last CS table for these ‘tiny’ frequencies (where the ‘right’ answer is to output 0). It turns out that it’s enough to ensure this failure probability is at most $1/poly(\log n)$, so using $O(\log \log n)$smaller CS tables in the filtering step suffices and gives a clean way to prove our improved bounds. It could be true that much fewer CS tables suffice for the filtering step, and this is an interesting question for future research.
>
> >Could you also discuss whether your results hold (strengthen or weaken) for more general power laws where f_i ~ (1/i)^p (or log-normal) beyond f_i ~ 1/i Zipf similarly to Du, Elbert, Franklyn Wang, and Michael Mitzenmacher. "Putting the “Learning." ICML, 2021? Probably it's best worked out and discussed after lines 222-223
>
> We did indeed consider questions of this type and for the first four algorithms of Table 1 (for power law distributions), we do have (nearly) tight bounds. For the remaining two algorithms, we have some partial analysis, which needs some polishing. We will consider including these bounds in the final version of the paper. It is an interesting direction for future work to consider other distributions like log-normal as you suggested.
>
> >Could you also measure and disclose the power law exponent for the CAIDA and AOL datasets?
>
> The frequency plots of both CAIDA and AOL datasets are given in Hsu et al and we will provide a reference to their discussion on power law parameters of these datasets.
>
> >Figures 2-5: Could you use the same color for best Our (C=..) line in the left and right sub-plots?
>
> Thank you for the suggestion. We will update the figure.
>
> >Lines 249-250: three columns and varying number of rows -> 3 rows and varying number of columns (typo).
>
> Thank you for pointing out the typo. It will be fixed in the updated version.

---

> > ### Comment · Reviewer_yxpU · 2023-08-21
> >
> > Thanks for the detailed reply, convincing explanations, and sharing further work in progress results. I'll update my final review accordingly.

---

### Decision · Program_Chairs · 2023-09-21

**Decision:**

Accept (spotlight)

**Comment:**

The reviewers unanimously agreed that this paper was of interest as it gives several compelling results including tight bounds for frequency estimation with count-min and count-sketch for Zipfian distributions, a simple new approach based on truncating small frequency estimates, which gives stronger error bounds, and a learning augmented version of that approach with even tighter error bounds. The paper is clearly above the bar and should be of general interest to those working on learning augmented algorithms and streaming algorithms.